# *Icerya purchasi* Maskell (Hemiptera: Monophlebidae) Control Using Low Carbon Footprint Oligonucleotide Insecticides

**DOI:** 10.3390/ijms241411650

**Published:** 2023-07-19

**Authors:** Nikita V. Gal’chinsky, Ekaterina V. Yatskova, Ilya A. Novikov, Refat Z. Useinov, Nanan J. Kouakou, Kra F. Kouame, Kouadio D. Kra, Alexander K. Sharmagiy, Yuri V. Plugatar, Kateryna V. Laikova, Volodymyr V. Oberemok

**Affiliations:** 1Department of Molecular Genetics and Biotechnologies, Institute of Biochemical Technologies, Ecology and Pharmacy, V.I. Vernadsky Crimean Federal University, Simferopol 295007, Crimea; pcr.product@gmail.com (N.V.G.);; 2Laboratory of Entomology and Phytopathology, Dendrology and Landscape Architecture, Nikita Botanical Gardens—National Scientific Centre of the Russian Academy of Sciences, Yalta 298648, Crimea; 3Centre National de Floristique, Université Félix Houphouët-Boigny, Abidjan 01 BP V 34, Côte d’Ivoire; 4Biology Laboratory and Animal Cytology, Université Nangui Abrogoua, Abidjan 02 BP 801, Côte d’Ivoire; 5Department of Natural Ecosystems, Nikita Botanical Garden—National Scientific Centre of the Russian Academy of Sciences, Yalta 298648, Crimea; 6Department of Biochemistry, S.I. Georgievsky Medical Academy, V.I. Vernadsky Crimean Federal University, Simferopol 295015, Crimea

**Keywords:** *Icerya purchasi* Maskell, green oligonucleotide insecticides, DNA insecticides, olinscides, climate change, carbon footprint, pH leaves, MALDI-TOF, insect invasions

## Abstract

Climate change creates favourable conditions for the growth of insect populations. Today, the world is seeing an increase in the number of insect pest infestations associated with a long-term increase in the average temperature of climatic systems. For example, local invasions of *Icerya purchasi* Maskell, a citrus pest recognized worldwide, have increased in size and number in recent years. Controlling this pest is complicated because not all chemical insecticides are effective, and their use is undesirable since citrus fruit is used for food and chemical agents cumulatively harm human health. In this article, we demonstrated for the first time the successful use of a short single-stranded fragment of the 28S ribosomal RNA gene called “oligoICER-11” to control cottony cushion scale, and we propose the use of green oligonucleotide insecticides with a low carbon footprint for large-scale implementation in agriculture and forestry. Using the contact oligonucleotide insecticide oligoICER-11 at a concentration of 100 ng/μL on *I. purchasi* larvae resulted in a mortality of 70.55 ± 0.77% within 10 days. Thus, climate change is driving the need in both agriculture and forestry for oligonucleotide insecticides (DNA insecticides, olinscides): safe, effective, affordable insecticides with a low carbon footprint and long operational life.

## 1. Introduction

Increased global surface temperature because of climate change affects crop insect pest populations in several complex ways, including increasing the number of generations per year and the risk of invasion by migrant pests [1,2,3]. In recent years, the impact of climate change on pests and their host species has been significant [4] because insects are poikilothermic organisms, meaning that their body temperatures depend on the temperature of the environment. Thus, temperature is probably the most important environmental factor affecting insect behaviour, distribution, development, and reproduction [5]. Therefore, it is likely that the main drivers of climate change (increased atmospheric CO_2_, increased temperatures, and decreased soil moisture) could significantly affect the population dynamics of insect pests and thus the percentage of crop losses [6]. Climate change favours the growth and spread of most pest species to new geographic regions and facilitates their movement from one region to another.

Human activity and globalization also contribute to the speed of biologic invasion, leading to the homogenization of biomes and the loss of biodiversity [7]. Invasive species cause harm to local ecosystems in several ways: they compete with, and may overrun, native species, and endanger local species by interfering with human management systems, such as those related to agriculture, animal health, and forestry [8,9,10,11]. Despite efforts to prevent insect pest invasions, the rate of invasions over the past few decades has risen steadily and remained at a high level due in large part to global trade in host plants and their spread [12,13]. The global cost associated with invasive insects has been estimated at USD 77 billion per year, equivalent to the combined cost of all goods and services including healthcare [14].

Greenhouse gas emissions from the application of pesticides and other chemicals constitute an environmental cost that has gone largely unrecognized [15]. Recent findings show that these human-related activities have contributed tremendously to global warming [16], primarily carbon dioxide (CO_2_), methane (CH_4_), and nitrous oxide (N_2_O) [15,17].

Among invasive species, insect pests with a large number of generations per year are of particular concern for agriculture. One such pest is *Icerya purchasi* Maskell (Hemiptera: Monophlebidae). *Icerya purchasi*, also known as the cottony cushion scale, is a cosmopolitan pest native to Australia and New Zealand and is known to have affected over 200 different plant species [18]. It is pest in several ornamentals and crops, such as *Citrus reticulata* Blanco, *Artocarpus heterophyllus* Lam., *Magnolia denudata* Desr., and *Ficus altissima* Blume [19]. Most damage occurs when early immature stages of the scale feed on the leaves, where they settle in rows along the midribs and veins, and on the smaller twigs. It causes decreased tree vitality, fruit drop, and defoliation [20]. This pest has also been introduced accidentally and is now found in 126 countries [21]. *Icerya purchasi* was introduced to California (U.S.) on *Acacia latifolia* Benth. around 1868 or 1869 and has since caused enormous damage to citrus groves in the south. It has also damaged *Rosmarinus officinalis* L. in Greece and *Pittosporum tobira* Thunb. in China [19,22,23]. The uncontrolled infestation of cottony cushion scale has had a severe effect on the pomiculture and horticulture industries and on the endemic fauna of the Galapagos Islands [24]. Moreover, the damage it causes to host plants leads to the extinction of Lepidoptera species that feed on them, such as *Semiothisa cerussata* Herbulot, *Platyptilia vilema* B. Landry, and *Tebenna galapagoensis* Heppner & Landry [25,26]. In Turkey, *I. purchasi* has caused extensive damage to cherry laurel (*Prunus laurocerasus* L.) orchards located in the Black Sea region [27] and mimosa plants (*Acacia dealbata* Link) in Artvin [28].

Organophosphates and petroleum oils are used to control this pest, and although buprofezin is effective on young nymphs, it fails to affect the adults [28]. At the same time, the demand for new insecticidal products has skyrocketed as insect pests have become more resistant to pesticides [29]. While the use of a predator species, *Rodolia cardinalis* Mulsant, has shown considerable potential in the control of cottony cushion scale populations [24,26,27,28], the activity of natural enemies is diminished by the blind use of broad-spectrum insecticides, since this kills both predator and prey; in addition, these insecticides have adverse effects on the environment [28,30,31]. When attempting to control insect pests for which chemical control is contraindicated, especially those for which natural predators are sparse, it is important to use effective biopesticides with a low risk of causing resistance and few if any toxic effects on the environment and human health [32].

As a solution, we propose the use of low-carbon-footprint oligonucleotide insecticides (DNA insecticides, olinscides) [33,34] for insect pest control. Modern solid-phase synthesis of oligonucleotide insecticides on DNA synthesizers using phosphoramidites does not lead to the accumulation of greenhouse gases such as nitrogen oxide, ozone, methane, or carbon dioxide. DNA synthesis occurs in an airless environment in an acetonitrile solution using catalysts. The main substances used for oligonucleotide synthesis are amidites, tetrazole, 1-methylimidazole, triethylamine, acetic or propionic anhydride, pyridine, iodine, acetic acid, trichloroacetic acid, dichloromethane, and acetonitrile. Compared to the chemical insecticide used to control *I. purchasi*, an oligonucleotide insecticide does not have a carbon footprint, although there could be a minimal amount in some cases (Table 1).

For the last three years, our research team has confirmed many promising aspects of insecticides developed from antisense oligonucleotides against scale insects of the suborder Sternorrhyncha (Hemiptera) [34]. Resistance to an antisense oligonucleotide corresponding to a highly conserved region of a gene develops slowly; for this reason, it is hard to ignore their enormous possible benefits. Insecticide resistance can be slowed by ‘basing’ DNA insecticides on very conservative regions of functionally important genes such as those that encode ribosomal RNA. This approach is of immense value, and further developments in this field may lead to safer, less expensive forestry and agriculture sustained by DNA insecticides. In recent articles in the *Insects and International Journal of Molecular Sciences* discussing our work with scale insects, we were the first to show that green oligonucleotide insecticides are highly effective against armored and soft scale insects, including *Unaspis euonymi* Comstock, *Dynaspidiotus britannicus* Newstead, *Ceroplastes japonicus* Green and *Coccus hesperidum* L. Therefore, these insecticides, based on an antisense fragment of the 28S ribosomal RNA gene of these insect pests, have the potential to replace many modern non-selective chemical insecticides thereby reducing the ecotoxicological burden on ecosystems [34,36,37,38].

In this study, to control the serious pest *I. purchasi* we pioneered the use of a low footprint oligonucleotide insecticide (oligoICER-11) as an alternative to neonicotinoids (e.g., thiamethoxam), the production of which is accompanied by the substantial release of CO_2_ [39].

## 2. Results

### 2.1. Search for the oligoICER-11 Target mRNA

To evaluate the possibility that the action of the oligoICER-11 occurred through the mechanisms characteristic for antisense RNase H-dependent oligonucleotides [33,40], we used PCR with 28S rRNA gene specific primers and revealed one specific fragment of cDNA derived from the total mRNA of the *I. purchasi*. This fragment from each insect DNA spectrum was chosen for DNA purification and further DNA sequencing. The sequenced cDNA fragment had a high cover by query (100%) in genetic database GenBank (*I. purchasi* 28S large subunit ribosomal RNA gene partial sequence; https://www.ncbi.nlm.nih.gov/nuccore/AY427432.1, accessed on 18 June 2023) with the sequenced 28S rRNA gene fragment (Figure 1), confirming that in experimental of larvae we used specific antisense oligonucleotide insecticide (range: 330 to 320, oligoICER-11) to target mRNA of a homologous 28S rRNA gene of the *I. purchasi*.

### 2.2. Mortality of I. purhcasi after Treatment with oligoICER-11 in the Natural Habitat

In *I. purchasi* larvae grown in a natural habitat, insect mortality increased significantly on the third day after treatment in the oligoICER-11 group (χ^2^ = 53.567, *p* < 0.01, *N* = 724, df = 1) compared with the mortality of larvae in the control (water-treated) group. In the groups treated with water, thiamethoxam, and oligoICER-11, we observed larval deaths of 14.02, 28.67, and 42.92%, respectively (Table 2).

On the tenth day after treatment, we observed a statistically significant increase in insect mortality caused by oligoICER-11 compared to that in the control group (χ^2^ = 96.464, *p* < 0.01, *N* = 545, df = 1). Among the three treatments, dead insects were found in the following percentage: 23.7% (Control), 35.33% (Thiamethoxam), and 70.55% (oligoICER-11). On the other hand, thiamethoxam compared to the control group showed only a moderate insecticidal effect on the third (χ^2^ = 19.785, *p* < 0.01, *N* = 631, df = 1), and tenth (χ^2^ = 3.151, *p* > 0.05, *N* = 634, df = 1) days. Thus, for the first time, we were able to significantly increase mortality larvae in natural habitat grown, using contact olinscide (oligoICER-11).

### 2.3. Biodegradability of Oligonucleotides with the Participation of Tissue Deoxyribonucleases

The constant use of chemical insecticides in agriculture and forestry poisons plants and animals at all trophic levels of aquatic and terrestrial ecosystems, where for most chemical agents (xenobiotics) there are no enzymes to catalyze their rapid decomposition [36]. It should be noted that when pesticides are applied to a particular area or plant, they inevitably decompose into new chemical compounds called metabolites and are involved in the processes of bioaccumulation and biotransformation [41]. Pesticides and their metabolites are transferred from target to non-target areas through adsorption, leaching, evaporation or runoff [42]. A characteristic feature of many pesticides is that they accumulate in living organisms, resulting in an increase of up to 20 times as the insecticide moves along the food chain [41]. In our previous work, we showed that deoxyribonucleases, which are present in the cells of the gypsy moth (*Lymantria dispar* L.), colorado potato beetle (*Leptinotarsa decemlineata* Say) and their host plants (*Quercus pubescens* Willd., and *Solanum tuberosum* L.), have a high specificity for oligonucleotide insecticides and ensure their degradation upon interaction with them [43,44]. However, it was important to investigate the activity of intracellular nucleases like *I. purchasi*, and *P. tobira*, the main host plant of the cottony cushion scale in the Crimea, and to evaluate the half-life and biodegradability of olinscides in target and non-target organisms (Figure 2).

Also, to evaluate any possible negative effects of oligoICER-11 insecticide on the plant, we measured the pH of the leaves (Table 3). No significant difference was found, confirming the environmental safety of oligoICER-11 insecticide (*p* > 0.05).

### 2.4. The oligoICER-11 Olinscide Significantly Decreases the Concentration of the 28S rRNA in I. purchasi Larvae

We evaluated the specificity of action of the olinscide oligoICER-11 by analyzing the expression of 28S rRNA. A decrease in the expression of action of the target gene is the gold standard for proof of specificity of action [41] for antisense oligonucleotides. The concentration of the 28S rRNA in oligoICER-11-treated insects was lower (1.4-fold) compared with that of the controls on the third day, and significantly lower (6.4-fold) compared with that of the controls on the seventh, as shown in Figure 3.

This provided evidence that the target 28S rRNA was degraded and that the oligoICER-11 fragment decreased its concentration as an antisense RNase H-dependent oligonucleotide [33,40]. The 28S and 5.8S rRNAs constituted 85–90% of the total cellular RNA and were very useful as internal controls [45]. Thus, in this experiment, the green olinscide oligoICER-11 recruiting RNase H was successfully applied to *I. purchasi* for the first time.

## 3. Materials and Methods

### 3.1. Origin of I. purchasi

After locating and identifying *I. purchasi* on *P. tobira* in the Nikita Botanical Garden (Yalta, Crimea) (Figure 4), we conducted three independent field experiments between October 2020 and February 2021.

### 3.2. Applied DNA Sequence (Olinscide oligoICER-11)

We used an algorithm of the web application BLAST Genomes (https://blast.ncbi.nlm.nih.gov/Blast.cgi, accessed on 18 June 2023) to choose the oligoICER-11 (5′-ACA CCG ACG AC-3′; GenBank: AY427432.1) sequence. The sequence was synthesized using an ASM 800E DNA synthesizer (BIOSSET, Novosibirsk, Russia). It was then used as a contact olinscide dissolved in a concentration 100 ng/µL in nuclease-free water and then applied using a hand sprayer to *P. tobira* leaves (mg of olinscide per m^2^ of leaves). The hand-sprayer volume of 10 mL was used per m^2^. A water-treated group was used as a control. During three independent experiments, around 3200 insects of first and second instars were treated, which were included in the statistical calculations to assess the survival rate. Insects were counted for each replication of each variant of the experiment on 40 *P. tobira* leaves. Mortality was calculated by dividing the number of dead insects by the total number on the 40 leaves and multiplying by 100.

### 3.3. Neonicotinoid Insecticide Treatment

To compare the effectiveness of the olinscide oligoICER-11 on *I. purchasi* with that of a commercially available chemical insecticide, the neonicotinoid insecticide Thiamethoxam (Syngenta, Austria) was used as a standard (active ingredient is thiamethoxam, concentration of 0.8 g/L). A hand-sprayer volume of 10 mL was used per m^2^.

### 3.4. Assessment of the Quality of Oligonucleotide Synthesis by the MALDI-TOF Method

The quality evaluation of synthesized oligonucleotides was determined using a BactoSCREEN analyzer based on a MALDI-TOF mass spectrometer (Litech, Moscow, Russia). The mass-to-charge ratio (*m*/*z*) of oligonucleotides was measured as positive ions with 3-hydroxypicolinic acid as a matrix on a LaserToFLT2 Plus device (UK) in a ratio of 2:1. The theoretical *m*/*z* ratio was calculated using ChemDraw 18.0 software (ChemDraw, CambridgeSoft, Cambridge, MA, USA) and differed by no more than 10 units (Table 4).

### 3.5. Search for I. purchasi 28S rRNA Fragment, Complementary of the oligoICER-11 Sequence

Total mRNA was extracted from larvae using an RNA Extract kit (Evrogen, Moscow, Russia), following the manufacturer’s protocols. First strand cDNA synthesis was performed using a MMLV RT kit (Evrogen), following the manufacturer’s protocols. Primers, forward 5′-AGG ATT CAC ACG GTG GAG TC-3′, and reverse 5′-GCA AGT GCA CAA CTT GAA CG-3′, were used for quantitative real-time PCR studies and amplification with gene specific primers. PCR reactions were performed on 2 µL of cDNA using 7 µL FastStart SYBR Green Master Mix (Roche, Basel, Switzerland), 2 µL ddH_2_O (Roche, Basel, Switzerland) and 0.5 µL (80 ng/µL) of each primer. DNA was initially denatured for 4 min at 95 °C, followed by 30 cycles of 1 min of denaturation at 94 °C, 1 min of hybridization at 56 °C, and 1 min of elongation at 72 °C, followed by a final elongation step at 72 °C for 7 min. PCR products from the larvae were purified using the Cleanup S-Cap (Evrogen, Moscow, Russia) and the sequencing polymerase reaction was performed with BigDye Terminator v 3.1 Cycle Sequencing RR-100 (Applied Biosystems, Vilnius, Lithuania). Polymerase reactions were performed with 2 µL purified DNA and 2 µL of primers (12.8 ng/µL). DNA was initially denatured for 1 min at 96 °C, followed by 30 cycles of 10 s of denaturation at 96 °C, 5 s of hybridization at 50 °C, and 4 min of elongation at 60 °C. Amplicons were sequenced in both directions with a capillary DNA sequencer (NANOPHOR-05, Syntol, Russia Applied). DNA sequences were analyzed using BLAST [46] and ClustalW 2.0.3 programs [47].

### 3.6. Quantification of I. purchasi 28S rRNA Gene Concentration

Total RNA was isolated from larvae *I. purchasi* using ExtractRNA Reagent (Evrogen, Moscow, Russia) according to the manufacturer’s instructions. Three independent extractions were carried out to produce the replicates for each treatment. For each extraction (25 mg larvae) were used for each group (Control, Thiamethoxam, oligoICER-11). The quality and concentration of the extracted total RNA was assessed using a NanoDropTM Lite spectrophotometer (Thermo Fisher Scientific, Waltham, MA, USA), and 1.8% agarose gel was used to run electrophoresis in TBE (Tris-borate-EDTA) buffer (10 V/cm) for 40 min. The quantity, intensity, and pattern of RNA bands were equal in all experimental groups, confirming the quality and reproducibility of RNA extraction from the insect material.

For reverse transcription, the total RNA (0.1 µg) was annealed with a reverse primer (5′-GCA AGT GCA CAA CTT GAA CG-3′) and analyzed using a MMLV Reverse Transcriptase kit (Evrogen, Moscow, Russia) according to the manufacturer’s instructions. The reaction was conducted at 40 °C for 60 min using a LightCycler^®^ 96 Real-Time PCR System (Roche, Basel, Switzerland). For each sample, a 0.5 µL aliquot of the obtained cDNA were used for each PCR reaction. Primers, forward 5′-AGG ATT CAC ACG GTG GAG TC-3′ and reverse 5′-GCA AGT GCA CAA CTT GAA CG-3′ for *I. purchasi*, were used for quantitative real-time PCR studies and amplification with gene specific primers to quantify the *I. purchasi* 28S rRNA. The FastStart SYBR Green Master Mix (Roche, Basel, Switzerland) was used according to the manufacturer’s instructions. The following procedure: 10 min initial denaturation at 95 °C, followed by 30 cycles with 10 s denaturation at 95 °C, 15 s annealing 56 °C, and 20 s elongation at 72 °C was used for amplification on a Light-Cycler^®^ 96 Real-Time PCR System (Roche, Basel, Switzerland). PCR was repeated in triplicate for each condition. Finally, to estimate the specificity of amplification and presence of additional products, all PCR products were melted.

### 3.7. DNA Nuclease Activity Analyses

The activity of intercellular DNA nucleases in tissue homogenates of target (*I. purchasi* larvae) and non-target organisms (*D. melanogaster* Meigen larvae, *P. tobira* leaves) was analyzed. A 5 mg aliquot of a target tissue was ground in 10 μL of water and then 10 μL of a target olinscide oligoICER-11 at a concentration of 100 ng/μL was added. Solutions were incubated for 0.3, 1, and 24 h. Finally, the solutions were centrifuged for 1 min at 12.000× *g*. The resulting supernatants underwent electrophoresis on 1.8% agarose gel with standard TBE buffer and ethidium bromide (10–15 μL at a concentration of 10 mg/mL per 55 mL of 1.8% agarose gel) as a nucleic acid stain.

### 3.8. Statistical Analyses

For statistical analysis to evaluate significance of the difference between control and experimental group on the third, seventh, and tenth days, the standard error of the mean (SE) was determined and evaluated using the Student’s *t*-test. The non-parametric Pearson’s chi-squared test (χ^2^) with Yates’s correction was used to evaluate the significant difference between the groups’ means. All above-mentioned calculations were made with Prism 9 software (GraphPad Software Inc., San Diego, CA, USA).

## 4. Discussion

Many insect pest management tactics that reduce the application of chemical insecticides also have the potential to further reduce greenhouse gas emissions [15]. One solution to reducing emissions in modern insect pest control efforts, in our opinion, is the use of the green olinscide we developed, which has a low carbon footprint.

The use of olinscide could also resolve, or at least improve upon, the important problem of insecticide resistance by slowing it down. The use of short single-stranded fragments of highly conserved segments of insect host genes should slow development of resistance to these insecticides because research has shown that the potential mutations responsible for changes to the target genes occur at a very low rate in the conserved parts of the genes. Thus, even if we cannot halt the genetic processes leading to insecticide resistance, we may be able to slow its emergence using DNA insecticides based on very conservative regions of functionally important genes, such as ribosomal (5.8S, and 28S rRNA) genes [36,37,44].

Moreover, our previous studies of the effect of green oligonucleotide insecticides on the biochemical parameters of *Quercus robur* L., *Malus domestica* Bokh [48], *Triticum aestivum* L. [49] *Manduca sexta* L., *Agrotis ipsilon* Hufnagel [50], and *Galleria mellonella* L. [44] showed their safety when applied to these non-target organisms. One of the most important aspects of insect pest control involves ensuring that a significant amount of the preparation enters the environment, which is frequently a large area that may feature difficult terrain. Using unmodified antisense oligonucleotides as natural oligomers seems to be the safest way to do this, since cells contain ubiquitous nucleases that can neutralize them [44].

Based on the above, the use of short, unmodified oligonucleotides as insecticides appears to be an attractive alternative since they work selectively, are subject to natural biodegradation unlike most chemical insecticides, and the commercial synthesis of olinscide in vitro is becoming more and more affordable [48]. Today, the method most widely used in the production of oligonucleotides is automatic solid-phase phosphoramidite synthesis [51,52]. However, solid-phase synthesis is inefficient, consuming on average several times more than the required number of reagents to create oligonucleotide chains [53]. A rational alternative could be liquid-phase synthesis, which can significantly reduce the cost by removing the need for CPG, the most expensive component of solid-phase synthesis (although low-cost controlled pore glass can be used for some DNA synthesizers) and removing the automatic synthesizers by transferring the synthesis to chemical reactors [54,55,56,57]. For this experiment, we found that creating the olinscide oligoICER-11 in a ready-to-use-preparation for *I. purchasi* control using solid-phase synthesis on automatic DNA synthesizers with controlled pore glass as a solid support cost around USD 0.5 per mg, or USD 0.5 per m^2^ of plant leaves. Although it is not yet affordable in comparison with thiamethoxam (USD 0.05 per mg), it may compete with conventional insecticides if the cost of DNA synthesis is reduced.

## 5. Conclusions

Since the use of many modern chemical insecticides leads to an increase in near-Earth temperature, it is necessary to consider not only their affordability, selectivity in action, biodegradability, and effectiveness during preparation but also whether their production and use are accompanied by CO_2_ emissions. Based on this, it must be emphasized that the use of chemical insecticides contributes to global warming, and global warming contributes to the expansion of insects into new territories in which they previously settled, forming the cornerstone of this problem. Olinscides have affordability, selectivity in action, fast biodegradability, and a low carbon footprint. We elaborated olinscides as a part of simferogenomics (Greek: συμφερο—usefulness), which uses antisense oligonucleotides as a tool for the selective regulation of insect pests at organismal and supraorganismal levels, benefiting agriculture and forestry [34]. Application of the olinscide oligoICER-11 was highly effective against *I. purchasi*, while the thiamethoxam had only a moderate insecticidal effect. Thus, the high efficacy of olinscides, along with their low carbon footprint during production, make them attractive candidates for large-scale use to control insect pests without contributing to climate change.

## Figures and Tables

**Figure 1 ijms-24-11650-f001:**
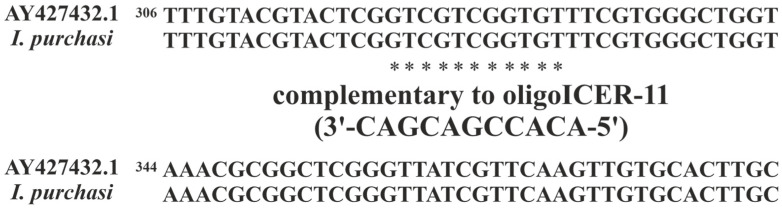
Alignment of the sequenced DNA fragments of *I. purchasi* (collected from nature) and *I. purchasi* 28S large subunit ribosomal RNA gene (from GenBank) fragments performed using ClustalW 2.0.3; complementary to oligoICER-11 sequences are marked with *.

**Figure 2 ijms-24-11650-f002:**
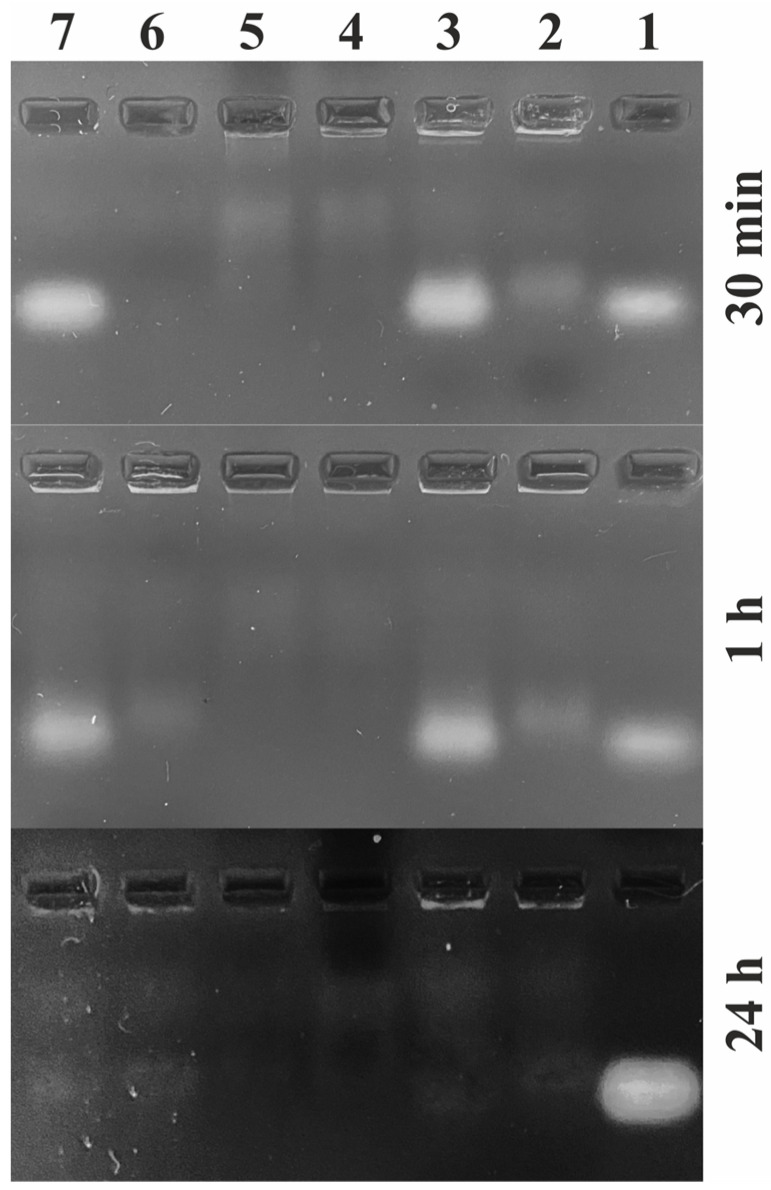
Electrophoregram (1.8% agarose gel) representing the activity of intracellular nucleases of *I. purchasi*, *P. tobira* and *D. melanogaster* after 0.3, 1, and 24 h at 27 °C: 1—control (10 μL of oligoICER-11 at a concentration of 100 ng/μL); 2—pure tissue homogenate *I. purchasi* larvae (1.5 mg of biomass per 10 μL of distilled water); 3—tissue homogenate *I. purchasi* larvae (1.5 mg of biomass per 10 μL of distilled water) + 10 μL of oligoICER-11 at a concentration of 100 ng/μL; 4—pure tissue homogenate *P. tobira* leaves (1.5 mg of biomass per 10 μL of distilled water); 5—tissue homogenate *P. tobira* leaves (1.5 mg of biomass per 10 μL of distilled water) + 10 μL of oligoICER-11 at a concentration of 100 ng/μL; 6—pure tissue homogenate *D. melanogaster* larvae (1.5 mg of biomass per 10 μL of distilled water); 7—tissue homogenate *D. melanogaster* larvae (1.5 mg of biomass per 10 μL of distilled water) + 10 μL of oligoICER-11 at a concentration of 100 ng/μL.

**Figure 3 ijms-24-11650-f003:**
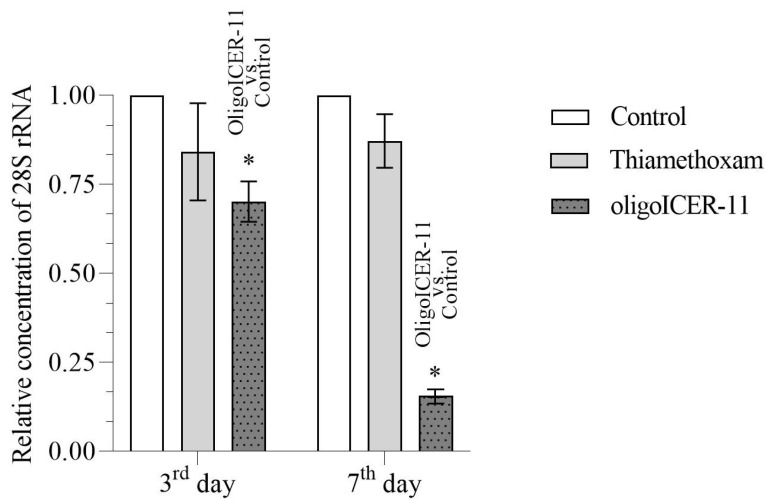
Relative concentration of 28S rRNA in *I. purchasi* after treatment with water, thiamethoxam, and oligoICER-11 on the third and seventh days. Data represent the mean and standard errors of ribosomal RNA concentrations for 3 replicates to the control (water-treated) group. Values for the control equal 1 (100%); * significant difference for *p* < 0.01 was calculated by the Student’s *t*-test.

**Figure 4 ijms-24-11650-f004:**
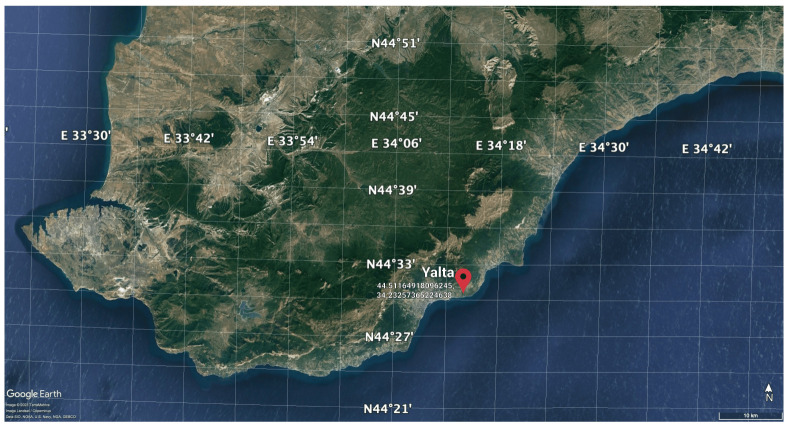
Location of the field experiments (Yalta, Crimea). The map is based on an image from ©Google, 2023.

**Table 1 ijms-24-11650-t001:** Comparison of greenhouse gas emissions associated with the manufacture of reagents for the synthesis of oligonucleotide insecticides and chlorothiazole neonicotinoide insecticides against *I. purchasi*.

Insecticides against *I. purchasi*	Corresponding Compound	Reagent and Solvents	t CO_2_/t Ratio	Reference
Synthesis of oligonucleotide insecticides (DNA insecticides, olinscides) on a solid-phase carrier on an automatic DNA synthesizer does not lead to greenhouse gas emissions				
Oligonucleotide insecticide (oligoICER-11)	Antisense oligonucleotide	Amidites, tetrazole, 1-methylimidazole, triethylamine, acetic or propionic anhydride, pyridine, iodine, acetic acid, trichloroacetic acid, dichloromethane, acetonitrile	~0	
Synthesis of frequently used chlorothiazole neonicotinoid insecticides results in greenhouse gas emissions				
Chlorothiazole (Thiamethoxam)	Thiamethoxam	S-methyl-N-nitroisothiourea, methylamine, N-methyl-N′-nitroguanidine, formaldehyde, formic acid, tetrahydro-1,3,5-oxadiazine, 2-chloro-5-chloromethylthiazole, dimethylformamide, potassium carbonate	0.351	[35]

**Table 2 ijms-24-11650-t002:** Mortality (%) of *I. purchasi* larvae.

Day	Control (Water)	Thiamethoxam	oligoICER-11
3rd	14.02 ± 2.79	28.67 ± 3.51 *	42.92 ± 2.25 *
7th	20.08 ± 6.25	33.67 ± 2.52 *	45.05 ± 4.74 *
10th	23.7 ± 8.87	35.33 ± 2.08	70.55 ± 0.77 *

Note: * significant difference for *p* < 0.01 by the Student’s *t*-test.

**Table 3 ijms-24-11650-t003:** The results of measuring the pH of leaves *P. tobira*.

Day	Control (Water)	oligoICER-11
1st	6.01 ± 0.06	6.02 ± 0.04
7th	6.04 ± 0.04	6.05 ± 0.04

**Table 4 ijms-24-11650-t004:** Results of the analysis of synthesized oligonucleotides by the MALDI-TOF method.

Oligonucleotide	Resulting *m*/*z*	Theoretical *m*/*z*
oligoICER-11	3295.72	3294.61
*I. purchasi* 28S rRNA gene forward primer	6189.09	6181.08
*I. purchasi* 28S rRNA gene reverse primer	6142.76	6134.09

## Data Availability

Not applicable.

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
