# Peer review of "Icerya purchasi Maskell (Hemiptera: Monophlebidae) Control Using Low Carbon Footprint Oligonucleotide Insecticides"

_ijms, 2023, doi:10.3390/ijms241411650_

Round 1

Reviewer 1 Report

The manuscript is well written and the research is presented clearly. The topic is current and will be of wide interest to readers in pest control and plant treatments for pests and pathogens.  The inclusion of price estimates will especially be of great interest to readers and growers for this emerging technology. Calculations per hectare are also very helpful in this type of research with oligonucleotides to aid estimates on potential commercial costs as the technology matures.  The authors do an excellent proof of concept presentation, and point out the primary applications, concerns and commercialization hurdles that still need to be overcome, while proposing a potential solution, in the use of liquid -phase synthesis technologies.  Rightly presented the authors point out that technology costs drop as it develops and mass production enters the equation.  

Author Response

Dear Reviewer,

please, see the attachment, thank you for your help and collaboration.

Kind regards, Dr. V. Oberemok and co-authors.

Reviewer 2 Report

Dear Authors, i believe that two things missing for your ms to be complete. a) Laboratory bio control with the three tretments b) better statistical analyses because is very poor.

See the pdf file for more comments

Author Response

(The authors gave the same response as above.)

Round 2

Reviewer 2 Report

Dear Authors please see the pdf file

Author Response

Dear Reviewer,

please, see the attachment.

Kind regards, Dr. V. Oberemok and co-authors
